# Understanding Quality of Pinot Noir Wine: Can Modelling and Machine Learning Pave the Way?

**DOI:** 10.3390/foods11193072

**Published:** 2022-10-03

**Authors:** Parul Tiwari, Piyush Bhardwaj, Sarawoot Somin, Wendy V. Parr, Roland Harrison, Don Kulasiri

**Affiliations:** 1Centre for Advanced Computational Solutions (C-fACS), Lincoln University, Christchurch 7647, New Zealand; 2Department of Wine, Food and Molecular Biosciences, Lincoln University, Christchurch 7647, New Zealand

**Keywords:** Pinot noir, wine quality, Buckingham Pi theorem, machine learning, sensory, chemistry

## Abstract

Wine research has as its core components the disciplines of sensory analysis, viticulture, and oenology. Wine quality is an important concept for each of these disciplines, as well as for both wine producers and consumers. Any technique that could help producers to understand the nature of wine quality and how consumers perceive it, will help them to design even more effective marketing strategies. However, predicting a wine’s quality presents wine science modelling with a real challenge. We used sample data from Pinot noir wines from different regions of New Zealand to develop a mathematical model that can predict wine quality, and applied dimensional analysis with the Buckingham Pi theorem to determine the mathematical relationship among different chemical and physiochemical compounds. This mathematical model used perceived wine quality indices investigated by wine experts and industry professionals. Afterwards, machine learning algorithms are applied to validate the relevant sensory and chemical concepts. Judgments of wine intrinsic attributes, including overall quality, were made by wine professionals to two sets of 18 Pinot noir wines from New Zealand. This study develops a conceptual and mathematical framework to predict wine quality, and then validated these using a large dataset with machine learning approaches. It is worth noting that the predicted wine quality indices are in good agreement with the wine experts’ perceived quality ratings.

## 1. Introduction

Wine tasting involves an interaction between an individual and a sample of wine. Everyone has a unique wine tasting experience, due to both physiological (e.g., sensitivity to a particular taste stimulus) and psychological processes, the latter including the individual’s unique, domain-specific experiential history [1]. Despite these established inter-individual differences, sufficient consensus exists amongst wine professionals and to a lesser degree, the consumer [2] for wine tasting studies to offer a reasonable degree of validity and reliability when investigating abstract wine attributes such as quality. For example, a Pinot noir wine that is perceived as being balanced and harmonious requires certain proportions of basic flavour and textural components [3]. Pinot noir is quite a costly wine, due to the high demand and challenging growing conditions for the grapes. In the world of wine, the complex flavour palette is unmatched, and it has a silky body and excellent ageing. Nonetheless, wine is a mixture of serendipity, culture and science, such that any innovative scientific approach that can integrate and model the relevant phenomena has the potential to pave the way to identify characteristics that individuals find pleasing (or dis-pleasing) to inform producers to both produce a more pleasing diversity of wines with appropriate characteristics for diverse consumers. Since the Second World War, the use of humans to assess the quality of food and beverage products has increasingly become the domain of science rather than industry [4]. An outcome of this is that nowadays, many food production companies make important decisions based on the scientific data produced by a panel of human assessors, rather than relying on the traditional model of an in-house taster [5]. Despite recent scientific interest, much of how people go about making sensory-based judgments of food products, i.e., the precise sensory and cognitive processes implicated, remains elusive. This is especially so with respect to complex food products such as wine.

The overarching hypothesis of this research is that wine quality, which is often perceived as a subjective and qualitative measure, could be related to the wine chemistry parameters through mathematical and machine learning approaches. In the initial phase of this research, a thorough sensory analysis was performed for two 18-wine sets of New Zealand Pinot noir wine (Experiment 1 and Experiment 2) to obtain the perceived wine quality indices. In the next phase, essential modulators are chosen from the available chemical data for set 1 on the basis of an extensive literature review and the threshold values of different chemical compounds present in the Pinot noir wine bottles. In the later stage of this phase, these essential modulators are modelled using the Buckingham Pi theorem that ultimately predicted the wine quality indices for individual bottles of Pinot noir. These quality indices are compared with the perceived wine quality indices obtained in the initial phase. The model’s computational framework follows a systematic approach, which is validated and verified using machine learning algorithms in the validation phase.

This mathematical pathway develops a scientific and artificial intelligence-informed decision-making process to predict quality indices for Pinot noir. Initially, we developed a reference model based on the properties of chemical compounds in the wine that characterise all of the features of a red wine, and in particular, Pinot noir for dataset 1. We do not have the characterisation of chemical compounds for another set of data (data3), but we do have the physiochemical data for both datasets. We used both physiochemical datasets as case study A and case study B to predict the wine quality, and to test the model’s validity for this kind of dataset. The predicted wine quality values are in good agreement with the perceived wine quality indices. The validation of the developed model and the results of two case studies are accomplished using machine learning algorithms.

### 1.1. Perception of Quality in Pinot Noir Wines

Recent years have seen the more abstract and elusive attributes of wine, including perceived quality, come under scientific scrutiny [3,6,7,8,9]. Both the perceived quality [3] and conceptualised quality [8] have been demonstrated to constitute multi-dimensional concepts involving intrinsic factors such as perceived balance and harmony, as well as extrinsic factors including wine price and the knowledge of the producer [3,10]. Further, overall quality appears to be a positive aspect of a wine for both wine professionals and wine consumers [8], with wine professionals also associating quality with the important, somewhat abstract wine attributes of complexity and varietal typicality [6].

In terms of the specific, intrinsic wine characteristics that drive perceived quality, some commonalities occur across wine types (e.g., in all red wines) and across grape varieties. For example, perceived bitterness has been shown to be associated with less preferred or lower quality wines across several red wine varieties, including New Zealand (NZ) Pinot noir [7] and Australian Cabernet Sauvignon and Shiraz/Syrah [11]. Such commonalities across wine varieties in terms of what constitutes quality appear rare, however, and the overall picture provided from the limited literature published to date suggests that the intrinsic characteristics important to wine quality differ across individual products, i.e., wine varieties. In the present study, our focus is limited to the fine red table wine Pinot noir.

*Vitis vinifera* L cv. Pinot noir produces table wines commanding amongst the highest prices paid for bottled wine anywhere in the world. The fine wines produced from this red grape express a combination of delicate [12], aromatic qualities and revered in-mouth attributes, the latter comprising a firmness or strength alongside a softness or silkiness [13,14,15,16]. These intrinsic qualities are assumed to have their source in the particular phenolic profile of the grape variety [3]. That is, Pinot noir grapes are typically reported as having a lower concentration of anthocyanins and tannins than many other well-known red varieties [17,18], this aspect of wine chemical composition giving rise to Pinot noir’s varietal typicality [6]. Inherent in the varietal nature of Pinot noir wine is a wine at the lighter end of the red wine colour spectrum, and a wine with a combination of specific floral and fruity aromas [6], along with tactile qualities, providing a power or strength combined with a soft, silky texture in the taster’s mouth [7].

In two previous studies, Parr and colleagues [6,7] investigated the specific wine characteristics perceived as being important to wine professionals’ judgments of Pinot noir overall quality. The studies employed the same 18 NZ Pinot noir wines and similar sensory methodologies, namely, a descriptive rating of experimenter-provided wine characteristics and sorting (classification) procedures (see [19] for the methodological and theoretical elaborations of these procedures). Experiment 1 [6] had as its focus predominantly aromatic attributes of the wines, whilst the second experiment’s focus was in-mouth wine attributes and in particular, wine texture. That is, the second study focused on wine attributes pertaining to taste (bitterness, sourness, and sweetness) and trigeminal stimulation, the latter being known as the mouthfeel of a wine [7]. Further, each study reported upon selected physicochemical aspects of the wines, with instrumental colour measuring being a focus in Experiment 1 and wine phenolic composition being the focus in Experiment 2. These studies were consistent in showing that the perceived quality differed significantly across the 18-wine sample set, an important prerequisite factor for differences in wine quality to be modelled via machine learning algorithms. The data from both experiments were consistent in showing that when a wine was perceived as being true to its variety, i.e., demonstrating high varietal typicality, it was also rated higher in perceived quality. Each study further demonstrated the specific sensory wine attributes, aromatic and textural, driving professional wine judgments of Pinot noir wine’s overall quality, along with several physicochemical correlates of the key sensory phenomena demonstrated. The data from these two studies have been employed as the sensory data in developing data driven and mathematical approaches to model wine quality.

### 1.2. Modelling Human Responses to Sensory Stimuli

Advances in technology over recent decades have permitted developments aimed at either replacing or modelling human responses to sensory stimuli. The purpose of such developments is to improve the reliability and/or validity of the data gathered in response to sensory stimulation, the desire for increased accuracy being driven by the known idiosyncratic nature of human perception, in particular with respect to the process of olfaction [4], olfaction being a process that is extremely important in wine sensory assessment. For example, electronic noses (E-noses), with their array of sensors, have been employed to detect a range of volatile qualities in a wine, in particular, those gases producing what typically are considered as the wine faults or off-notes [20,21].

In terms of attempts to model how the human sensory system interprets and appreciates a complex product such as wine, various technologies associated with neuroscience (e.g., EEG to measure cerebral electrical activity and cerebral imaging techniques) have been employed. In the limited research endeavours published to date [22], this approach has been aimed at providing more objective data and outcomes relative to those provided using a cognitive analysis of wine tasting phenomena [1]. That is, a cognitive analysis of wine-tasting data requires the use of intervening variables to define constructs such as expectations, cerebral representations, memories, and so forth, to interpret data from sensory-based judgments. Such neuroscience-based studies have been limited in their effectiveness in providing an understanding of the precise processes implicated in sensory judgments of wine. That is, whether a study participant can ‘taste’ a wine with any degree of ecological validity whilst lying prone in a scanning device used for cerebral imaging.

More recently, researchers have begun to develop and apply artificial intelligence and machine learning approaches to take some of the uncertainty out of wine production and wine assessment. Viticultural and oenological phenomena, including weather patterns, soil types, fruit ripeness, and wine classification have been of particular interest [23,24,25,26,27]. In terms of appreciating and judging the finished wine in the glass, limited studies have begun to appear that involve the use of machine learning algorithms. The study reported by [24] investigated an important red wine intrinsic attribute, namely, perceived astringency. Employing both sensory and wine chemical composition data in their machine learning model of wine astringency, the authors reported various aspects of wine polyphenolic composition that were important to perceived astringency. A recent study relevant to our study investigated the prediction of Pinot noir wine colour and other sensory descriptors from weather events and management practices [25] across nine vintages, although perceived wine quality was not an attribute that was assessed. The authors reported that the use of weather information and vineyard water-management data advantaged a model’s accuracy in predicting wine sensory profiles, including wine colour [28,29]. In our study, we employ computational and machine learning models to combine both sensory and selected physiochemical data on a set of NZ Pinot noir wines to predict wine professionals’ judgments of overall wine quality.

### 1.3. Chemical and Physiochemical Correlates of Perceived Quality

Wine descriptions outside the scientific domain can appear quite fanciful. Wine aroma often is given much weight in such descriptions, as it plays a vital role in defining a wine’s attributes, including its quality and varietal typicality [30]. In scientific analyses, the perceived aroma is typically defined as resulting from ortho-nasal and retro-nasal olfactory processes. In recent years, however, aroma has been argued as being inherently entwined with taste and trigeminal stimulation. As such, aroma is an integral part of a multi-sensory perceptual process [31] involving interactions with texture and taste. In turn, all these perceived wine qualities result from the viticultural and vinification practices applied to grapes, including fermentation temperatures and maturation processes. What is clear is that the intrinsic quality of an individual wine cannot be judged just by reading the bottle label.

Chemical compounds play a vital role in the composition and sensory expression of wines. This is especially so for wines and red wines, which tend to be more complex, but ultimately the chemical composition of any wine is what the taster perceives [32]. A good understanding of the interlinkage between the perceived wine quality and these chemical compounds can certainly help in the creation of a unique wine style. Modulating the significant characteristics of a wine, while maintaining the best viticultural, oenological, and management practices, is an ongoing challenge for the wine industry [30,33]. Due to their domain-specific experience, and their extensive knowledge of wine production processes and the chemical composition of various wine varieties, wine experts [34] tend to have a consensual perspective when judging wine, including its quality and complexity [2,34,35]. Research suggests that everyday consumers tend to associate wine quality with how much they enjoy a wine, along with extrinsic factors such as the wine’s price, its presentation, and the wine’s origin [8,10]. Consumers may struggle to judge a wine’s quality, due to the sheer number of available wines [36], although have some pre-conceived ideas about the type and nature of a particular wine, due to their prior experience with some wines [2].

A wine’s aromatic characteristics are associated with the volatile compounds [37] contained within the bottle. By themselves, volatile compounds cannot provide the wine with an aroma; it is often through their combinations, but not always, along with the human processes of sensation and perception, that one can detect the aromatic characteristics of an individual wine [38]. These compounds can saturate the wine with different kinds of aromas that are recognised as flavours such as fruity, smoky, herbaceous, coconut, green apple, and so forth. In wine, the aromatic compounds are related to diverse groups of chemicals. A few of them act as precursors for other groups. Some of the most important classes of volatile compounds present in wine, along with their subtypes, are aldehydes, higher alcohol, esters, terpenes, pyrazines and norisoprenoids [11,37,39]. The concentration of each of these compounds depends on a multitude of viticultural and oenological factors such as the grape variety, the soil in which the grapes are grown, the nutrients and water present in the grapes, method of grape harvesting, and the precise vinification processes, such as the type of maceration, the fermentation temperature and yeast species. Correlation maps among the various chemical compounds used in this study are available in the Appendix A.

Wine quality depends on wine chemistry, which involves numerous variables, or we can say the particular mix of chemical compounds that are present in the grapes, must and eventually, the finished wine. Let us consider a wine bottle containing different chemical compounds related to its aroma, mouthfeel (e.g., astringency or softness) and taste (e.g., bitterness) parameters, and define a state vector, $S:=\left(s_{1},s_{2},\ldots \ldots ,s_{n}\right)$ having n states. Each state $s_{i},\;i=1,2,\ldots ,n$ defines a different feature (say, herbal, floral or fruity) and contains many variables characterising these features. For the sake of understanding, say, $s_{i}$ represents the herbal aroma of the wine described by a few chemical compounds (say, Hexan-1-ol, (E)-Hex-3-en-1-ol, and Heptan-1-ol). The combination of these compounds for each state $s_{i}$ contributes to wine quality and plays a crucial role in defining good quality wine [40]. Though each variable may differ in terms of the percentage weightage of their contribution, some of them work as the driving force. We identified these driving forces and developed a conceptual and mathematical framework that can predict wine quality.

### 1.4. Computational Modelling of Perceived Quality

To develop the mathematical framework of the proposed model, we employed a semi-empirical approach based on dimensional analysis (DA). The DA approach is widely used in several branches of sciences and technology, including mathematics, physics, engineering and economics [41,42,43,44]. We chose this analytic approach because of its ubiquitous nature and its ability to solve complex problems over a wide range of disciplines [19]. DA is an approach that simplifies a physical problem using dimensional homogeneity. This analysis physically extracts meaningful relationships from the observed data. The role of DA is to restructure the set of original critical parameters consisting of true information into smaller sets of self-contained dimensionless parameters. These dimensionless parameters will themselves constitute a predictive model for the quantity of interest: in this case, the quality of Pinot noir wine.

### 1.5. Industry Relevance

A mathematical and machine learning model that has been trained to estimate wine quality is not subjective like a skilled human wine tasting is. Wine experts follow their own inclinations, whereas machine learning methods offer reliable forecasts in a more neutral manner. The right input data—labelled, of course—ensures the most accurately predicted outcomes, even while machine learning processes are directed by humans. Credit goes to machine learning models so that we can discover exactly what qualities a good wine has. In the wine sector, product quality certification is an expensive and time-consuming process for manufacturers. Machine learning is now a crucial technique, and it has automated the process of wine quality prediction for businesses.

## 2. Methods

### 2.1. Sensory Methods

#### 2.1.1. Participants

Twenty-two wine professionals (6 females and 16 males) with a mean age of 42.7 years (range: 33–62) participated in Experiment 1, and 17 (6 females and 11 males) with a mean age of 41.4 years (range: 24–62) participated in Experiment 2. All participants were experienced with the production and tasting of NZ Pinot noir wine, and were considered to be wine experts, as definedin [34]. They were primarily oenologists and winemakers; their overall expertise demonstrated in their mean number of years of wine industry experience was 18.2 years (range: 3–40 years) for those in Experiment 1, and 17.8 years (range: 6–40) for Experiment 2. The participants did not undergo any type of formal training prior to their participation in the current study, although several had attended prior research tastings in [35] and were familiar with the environmental controls, such as individual booths, and methodological requirements, such as the types of rating scales employed.

#### 2.1.2. Wines

Eighteen commercial Pinot noir wines from five NZ regions were employed, the same 18 wines in each of the two experiments (Table 1). Fifteen wines were from the 2016 vintage, and 3 from the 2013 vintage. Four producers had two wines each in the 18-wine sample set, and each of the other 10 wines was produced by a different company. Wines were selected as representatives of the major Pinot noir producing areas of NZ. As well, they spanned a range in terms of recommended retail price (RRP), price often linked anecdotally with quality. The details of the 18 wines can be seen in Table 1, along with several viniviticultural properties (e.g., wine region and method of production) that were used as controlling factors, i.e., to ensure the representative nature of the sample of wines selected. The viniviticultural factors were not treated as independent variables in the present study for valid reasons (e.g., low and unequal numbers in some cells). The two experiments were conducted 9 months apart so that the wines in Experiment 2 [7] had 9 months more bottle age than the same wines employed in Experiment 1 [6].

#### 2.1.3. Experimental Design

The design for both experiments was within-subject; this ensured that every participant evaluated each wine in each of the two sessions. In Experiment 1, Session 1 comprised a descriptive rating (DR) of the wines in clear glassware, while Session 2 involved the same wine assessment task, but with the wines in opaque (black) glasses. Nine participants undertook Session 1 first, followed by Session 2, while the remainder of the participants undertook Session 2 first, followed by Session 1. In Experiment 2, the two sessions involved a directed sorting task in Session 1 and a DR task in Session 2, with all participants undertaking the two sessions in the same order.

#### 2.1.4. Procedure

The procedural conditions were similar across the two experiments in terms of fundamental aspects. Both studies were conducted at the Marlborough Research Centre (MRC) in NZ, Experiment 1, in late 2018, and Experiment 2 in middle of 2019. Participants were welcomed to the MRC sensory facilities and seated in separate booths, with three to six people participating at any particular time. The environment of the facilities was controlled, as advised for sensory experimentation [45]. The requirements of the Lincoln University, NZ, Human Ethics’ Committee were adhered to. This included the provision of an information sheet describing what the participation involved, and a participant’s rights. Any queries raised by the participants were responded to by the experimenter, prior to participants signing consent forms in keeping with the ethical requirements. The sole, task-specific information provided to participants was that they would taste and make judgments about the 18 wines, and that all wines were Pinot noir. The participants were not subjected to any form of training (e.g., the provision of reference standards or a lexicon) prior to their participation.

The wines were served at room temperature (approximately 22 °C). For each day that an experimental session occurred, a new bottle of every wine was opened, and the opened wines were checked for faults by two or three experienced wine professionals. The wine samples for each task comprised 50 mL volumes, with a new sample poured for each session’s chemosensory evaluations. In Experiment 1, Session 1, the wine samples were served in clear Spiegelau tasting glasses (Spiegelau # 440 01 31, Spiegelau, Germany), while in Session 2, the wines were served in black Spiegelau tasting glasses (Spiegelau # 440 85 31). In Experiment 2, all wine samples were served in the black Spiegelau tasting glasses. The glasses were coded with 3-digit numbers and were covered with plastic Petri dishes. In order to limit carry over effects and memory biases, all wine samples were presented in a different order that was specific to each participant within each session, between sessions and between experiments, according to a Williams Latin square arrangement generated using FIZZ software (Biosystemes, Courtenon, France). Water and plain water crackers were available throughout each session.

For both experiments, the participants undertook the two sessions on the same day, the sessions being separated by an interval of approximately 20 min. Instructions to the participants included that they were to evaluate each wine in the order presented, and that all wine was to be expectorated (i.e., not swallowed). In Experiment 1, Sessions 1 and 2 were identical procedurally, apart from glass colour and wine order. Each participant first rated the wines on the 20 experimenter-provided descriptors reported in Table 2, column 1.

The 20 descriptors were the most appropriate attributes for the sensory characterisation of NZ Pinot noir wines [3,14]. The descriptors included wine characters that were assumed as being capable of assisting in differentiating wines of varying aromatic and phenolic compositions, and of differing degrees of perceived quality [3]. The attributes were scored on 10-point scales, with each scale’s anchors as described in Table 2, and were rated in the same order by each participant and in each condition. Overall quality was the first attribute rated, and varietal typicality was the last item rated by each participant. The data from the black-glass condition only constitute the Experiment 1 data reported in this article. Participants also completed a wine complexity questionnaire as their final task in Experiment 1, but those data are not reported here [6].

In Experiment 2, participants first undertook a directed, extended sorting task [7]. They were advised to evaluate each wine, with their focus being on in-mouth characteristics, avoiding purposeful olfaction (i.e., smelling of the wine). In-mouth characteristics were defined as tastes and mouthfeel characters (trigeminal effects), and it was explained that this did not include retronasal aromas. Participants were then advised that the wine evaluation task involved them sorting and categorising the 18 wines on the basis of the similarity of in-mouth perceptions (tastes and textural qualities), making as many groups as they found appropriate. The extended sorting task then required them to provide 1–2 descriptors to explain the criteria they had employed in their classification. The sorting task data are not included in the current article, but are described in [7].

Participants then took a 20-min break before their second session, which involved a DR task. They were provided with a new flight of the same 18 wines, and were asked to rate each one on the 17 descriptors in Table 2, column 3. The descriptors were selected to exemplify Pinot noir wine in-mouth characters as reported by wine critics [46]. For both global or higher-order terms (e.g., astringency and body), and more specific terms (grainy and drying/puckering), many of the latter, assumed to be sub-components of the global terms, were included for rating. Participants were reminded to focus on in-mouth characteristics when assessing a wine, avoiding purposeful olfaction. In keeping with Experiment 1, descriptors were to be rated on 10-point scales, in the order given by experimenters, and with overall quality being the first attribute to be rated, and varietal typicality being the last.

### 2.2. Mathematical Methods

#### 2.2.1. Identifying Essential Modulators

In the conceptual framework of developing a mathematical model for quality indices, we focused on essential modulators by keeping in mind the complex nature of Pinot noir wine (Figure 1).

The selection of these parameters plays a vital role in making accurate predictions about an individual wine’s quality, and requires careful study and technical knowledge of the chemical quantities that exist in the wine bottle. The more abstract concepts of wine balance, harmony of components, varietal typicality and so forth, are also major drivers of a wine’s perceived quality. As well, the mouthfeel or trigeminal sensations of astringency and softness are pivotal to the perceived quality (our sensory data demonstrate this emphatically), as well as the tastes of sourness and bitterness. We have selected modulators representing fruity, herbal, floral, woody, and other flavours, such as spicy.

We designated those modulators as fruity $\left(M_{1}\right)$, herbal $\left(M_{2}\right)$, floral $\left(M_{3}\right)$, woody $\left(M_{4}\right)$ and other flavours $\left(M_{5}\right)$. Each $M_{i};\;i=1,2,\ldots \ldots ,5$, is characterised by a number of chemical compounds. For example, ethyl butanoate, ethyl octanoate and ethyl hexanoate are the most important chemical compounds that are responsible for a fruity flavour. The selection of these chemical compounds was based on odour activity value OAV (OAV is the ratio of compound concentration to its odour detection threshold—ODT), an exhaustive literature review [14,30,31,32], experts’ opinion [3,10,47,48] and on the important aroma characteristics of Pinot noir wine. The important chemical compounds selected for these modulators are given in Table 3, and a detailed description of these modulators is explained in Section 3.2.

ODT and OAV are important when considering the potential contribution of volatile compounds to the aroma of the wine [14].

Many potent compounds have low odour detection thresholds, meaning that the presence of these compounds in wine can be detected at low concentration. Based on the experimental evidence, compounds with an OAV of greater than 1 are considered as high contributors to the wine aroma. However, in this study, we selected some compounds based on the experiments conducted in [14], even if they were not eligible based on the OAV and ODT. In total, five modulators were chosen for chemical data based on the above-mentioned criteria. The variables in $M_{1}$ are mainly responsible for the fruity aroma of the wine, and can be found in Pinot noir wine within a range of 18.5 to 874 (μg/L) and an ODT from 2 to 20 μg/L, together with an OAV from 0.92 to 447. It has also been suggested in [14] for some important compounds based on the correlation analysis to be obtained in the first essential modulator set. This modulator set contains volatile compounds from the ethyl ester group, such as ethyl octanoate, ethyl butanoate, and ethyl hexanoate. On the other hand, the $M_{2}$ group comprise a subset of the higher alcohols such as hexan-1-ol, (E)-hex-3-en-1-ol and heptan-1-ol, and they contribute to the herbal characteristics of the wine. Based on the correlation analysis conducted by Tomasino [14], heptan-1-ol is considered as being a necessary chemical compound due to its contribution to the wine aroma. In addition to that, $M_{3}$ contains β-damascenone, 2-phenylethan-1-ol and linalool, contributing to the floral characteristics of Pinot noir. According to the literature, β-damascenone in Pinot noir is found within a range of 0.2–102 μg/L, together with an ODT and OAV of 0.05 μg/L and 4–2030, respectively. Despite its low content, beta-Damascenone contributes significantly to the aroma of roses. On the other hand, 2-phenylethan-1-ol and linalool are mostly present in Pinot noir, with concentrations of 236–158,473 μg/L (ODT: 10,000–14,000 μg/L and OAV: 0.24–15.8) and 0.83–170 μg/L (ODT: 25 μg/L and OAV: 0.03–6.8), respectively. These two compounds are also considered as important, based on the correlation analysis, and they contribute to the aroma of Pinot noir with greater than 80% standardised weighting in one or more correlations. Moreover, eugenol, benzaldehyde and guaiacol were considered in modulator set $M_{4}$. The OAV and ODT values of guaiacol and eugenol were found in the ranges 0.32 to 14.4 and 5 to 10 μg/L, respectively. However, the selection of benzaldehyde was completely based on the findings in [14], where it was found to be important for wine aroma together with guaiacol and eugenol, and is a common chemical compound, along with phenol.

#### 2.2.2. Buckingham’s Pi Theorem for Predicting Wine Quality Indices

The roots of DA, as explained in Section 1.4, come from a century-old theorem known as the Buckingham Pi theorem [49]. DA empowers the conversion of physical variables into other various fundamental sets of measurement units called Pi-values [50]. In this theorem, a basic linear algebra approach is used to develop a well-organised mathematical framework to obtain dimensional scaling factors [51,52]. This analysis may not provide complete information about the inner connectivity of an investigated phenomenon [43]. However, it is an important and remarkably efficient way of understanding the behaviour of such a phenomenon for which no complete mathematical and physical description is available [42,44]. To our knowledge, no prior studies have used DA to develop a predictive model to establish the quality of Pinot noir and to determine essential modulators that are responsible for perceived good quality wine.

Thus, a brief overview of this approach is justified. In total, 18 samples of Pinot noir wine, selected from different wine regions in New Zealand, were used. According to this theorem, a quantity of interest (wine quality) or the dependent variable, say Q, is completely determined by a set of m independent quantities known as Pi-values, say $\pi_{1},\;\pi_{2},\ldots \ldots ,\pi_{m}$. These m quantities completely form an independent and dimensionless subset. We termed these quantities as essential modulators, as they represent the model’s degree of freedom.

Mathematically, for a given physical problem in which one dependent parameter Q is a function of the m independent variables:(1)$$Q=f\left(\pi_{1},\;\pi_{2},\ldots \ldots ,\pi_{m}\right)$$

Alternatively, we can write the equation as:(2)$$f\left(Q,\pi_{1},\;\pi_{2},\ldots \ldots ,\pi_{m}\right)=0$$

In the development of the current mathematical framework (Figure 2), this theorem serves as the theoretical basis of a modern experimental simulation method. A detailed explanation on the Buckingham Pi theorem is included in Appendix A.

It derives the implicit functional relationship between the chemical compounds involved in Pinot noir data through the dimensional relation, and then defines the specific proportional constant in the relation through theoretical analysis [42]. The output of this model (quality indices) is compared using sensory experimental data, and is validated through a machine learning model.

More importantly, this approach serves two purposes:Selecting important variables through DA before conducting further analysis saves time. It shows a clear direction in selecting relevant variables that ultimately improve the accuracy of the model in predicting wine quality.

### 2.3. Machine Learning Analysis

Machine learning analysis was conducted using chemical data (Table 4), and physiochemical data from sensory Experiment 1 and physiochemical data from sensory Experiments 2 and 3. Overall, we conducted three separate machine learning experiments with three different datasets (from here onwards, we refer to the chemical data from Experiment 1 as data1, the physiochemical data from sensory Experiment 2 as data2 and the physiochemical data from sensory Experiment 3 as data3). However, during machine learning analysis, we only considered the essential modulators as features for predicting wine quality. Data1 contains 14 features, while data2 and data3 contain 6 features. Prior to machine learning analysis, we performed the synthetic minority oversampling technique (SMOTE) [53] on all three datasets, due to the small number of samples in each dataset. Using SMOTE, we synthetically generated a total of 1000 samples in each dataset. Before implementing SMOTE, 18 samples from each dataset were divided into two sets, one set containing 12 samples, and another set containing 6 samples. Twelve samples in each set were used to generate new samples (test samples), and the remaining six samples (training samples) were used to evaluate machine learning models.

The SMOTE method is typically used to balance data by creating minority class samples that may be matched against the majority class [54]. The SMOTE algorithm utilises the KNN method to generate new samples by setting the minority class as set A. For each x∈A, k the nearest neighbours (x′) were attained by calculating the Euclidian distance between x and every other sample in A. Similarly, for each x∈A, N (the sampling rate) was randomly selected from its KNN to construct a new minority class set A_1_ equivalent to the majority class. The formula outlined below explains how we generated the new samples.
$$\;x^{\prime }=x+rand\left(0,1\right)*\left|x-x_{k}\right|$$

Here, rand(0,1) represents random numbers, with values between 0 and 1 [55].

Say, for example, if a dataset contains 1000 samples, 600 of which are red wine and 400 of which are white wine, white wine is the minority class while red wine is the dominant class. However, in our study, the situation is different: we received a total of 18 samples, all of which were Pinot noir wines. We therefore made a few assumptions to produce synthetic data. Let us understand it with an example of synthetic data generation using data1:

In the beginning, we created a dummy dataset with 1012 rows and 15 columns in the excel spreadsheet. These 14 columns provided chemical data and wine quality information. The first 12 rows of this dataset belonged to the original samples, and the remaining rows contained value 0, which we inserted manually. Now, we added another column to the data and named it as class column. The first 12 rows were considered as class 0, and the remaining rows containing value 0 were considered as class 1. Now, class 0 is the minority class because it contains less samples than class 1 (1000 samples). Next, we used this dataset for data generation using SMOTE. After applying SMOTE to the dummy dataset, we removed the class column and all the rows containing value 0. At this stage, we have a total of 1000 rows and 15 columns, in which the last column is related to the wine quality indices. At this stage, we have 1012 samples with 14 independent variables and 1 dependent variable (wine quality). This dataset was used in the training of the machine learning models.

Similar steps were used to generate synthetic data for data2 and data3. Descriptive statistics of data1, data2 and data3 are tabulated in Appendix A.

Now at this stage, we have three datasets (data1, data2 and data3) containing 1018 samples each. To predict the wine quality, we performed regression analysis using fully connected, feed forward deep neural networks (DNN) on all three datasets differently. DNN relies on deep learning strategies, which is a subset of machine learning widely used to perform analysis on complex data [56]. DNN is an artificial neural network containing multiple layers between the input and out layer. DNN models were trained using newly synthesised data (1000 samples), and tested on 18 original samples (out of which, 6 samples were not used in data generation). Before DNN analysis, we performed normalization on the dataset. Details on DNN are summarised in Appendix A.

Wine quality prediction using data1 was conducted by implementing a fully connected DNN with five layers. The layers in any model contain several nodes or neurons, an activation function such as rectified linear unit—RELU [57], a leaky rectified linear unit—LRELU, an exponential linear unit—ELU, tanh and more. Moreover, in the first layer, one should include the input dimension, and in this case, we had 14 input dimensions (essential modulator). We used a total of five layers, the first and second layers comprising 64 nodes with RELU and ELU activation functions, respectively. On the other hand, the third and fourth layers contained 32 and 16 nodes with ELU activation functions. The remaining layer was the output layer and provided wine quality information.

DNN also requires a type of optimiser such as Adam, RMSprop and more. For this study, we chose RMSprop [58]. During model training, the mean absolute error (MAE) was used as the evaluation metric with 2000 epochs. However, an early stopping strategy is also adopted, in which model will stop if it finds appropriate results at a low epoch. The selection of all these parameters was finalised using trial and error. Similar steps were implemented in the cases of data2 and data3. All the selected parameters are mentioned in Table 5.

### 2.4. Data Analysis

The sensory data included in this article involve descriptive ratings to the attribute overall quality in (i) the black glass condition of Experiment 1, and (ii) the DR condition (where only black tasting glasses were employed) in Experiment 2. Ratings to the descriptor overall quality were analysed through separate analyses of variance (ANOVA) for each experiment. A mixed effects model was fitted to the ratings of overall quality, with Wine as the fixed factor and Panelist as the random factor. Sensory models were fitted with R package lmerTest. Density plots for chemical compounds for dataset1 are shown in Figure 3.

## 3. Results

### 3.1. Sensory Results

The ANOVA results showed significant differences amongst the 18 wines for ratings of overall quality in both Experiment 1, *F* (17, 353 = 1.9, *p* = 0.015), and Experiment 2, *F* (17, 240 = 2.7, *p* < 0.001). Figure 4 shows the mean ratings of perceived quality to each of the 18 wines.

Results from the sensory experiments are summarised below:

Experiment 1, titled Perception of quality and complexity in wine and their links to varietal typicality (see [6] for a full report of this experiment):

Twenty-two wine professionals assessed 18 NZ Pinot noir wines (2013 and 2016 vintage) in both clear and opaque (black) glassware. Key results were:Visual influence (i.e., the glass-colour manipulation) was not a major driver of judgments of perceived quality, despite anecdotal evidence suggesting that it would be.Perceived quality differed significantly across the 18 wines, but wine region was not a significant factor (the within-region variability was too great to find the between-region significant variation with such a small # of wines in the sample set).

The perception of the quality in the wines was highly and positively associated with perceived varietal typicality, a concept that refers to whether a Pinot noir wine exemplifies a taster’s concept of what a Pinot noir wine should be like. (i.e., is the wine true to grape type?)The perception of quality was closely and positively associated with perceived complexity, and to a lesser degree, with the perceived familiarity of a wine, these two concepts have been shown in food science research to be important influencers of consumer preferences and behaviour.Key, specific drivers of Pinot noir quality were wine attributes of attractive fruit aromatics, expressiveness, overall structure, harmony and balance.

Experiment 2, titled Understanding in-mouth sensory phenomena important to perceived quality in NZ Pinot noir wines (see [7] for a full report of this experiment):

Seventeen wine professionals assessed the same 18 wines as employed in Experiment 1, in a tasting where they were instructed to focus on in-mouth wine attributes (tastes and trigeminal effects) rather than olfaction (aroma). The purpose of this was to assess the importance of mouthfeel qualities and the taste of bitterness to judgments of overall wine quality. Key results were:The 18 wines differed significantly on most of the in-mouth attributes assessed, including the perceived overall quality.Again, quality and varietal typicality appeared virtually synonymous concepts for the tasters.The major sensory dimension separating the wines was a tactile aspect, with wines judged as soft, gentle, smooth, silky, velvety and supple; as opposed to wines judged to be sour, bitter, coarse, rough, astringent and with harsh tannins (PCA output).A second important dimension was related to wine overall body, with attributes of weight, heaviness, density, fullness, roundness and volume opposing the descriptor ‘thin/watery’.Perceived overall quality was positively associated with the tactile attributes relating to the soft/smooth aspects, and negatively correlated with bitterness, astringency and harsh tannins.Overall, mouthfeel attributes important to quality were multi-dimensional, involving tactile (e.g., harsh/soft), body/weight and oiliness/viscosity dimensions.

Experiment 3 was titled Influence of provenance and vine yield on perceived quality of NZ Pinot noir wines. Wine professionals assessed a flight of 18 NZ Pinot noir wines (2019 and 2020 vintage) that differed from the wines employed in the prior two experiments. The results demonstrated that the 18 wines comprised a perceptibly diverse group of wines that differed significantly in terms of both perceived quality and perceived varietal typicality. The perceived wine quality ratings for Experiment 1 and Experiment 3 are shown in Figure 5.

### 3.2. Mathematical Results for Quality Indices (Selection of Optimal Essential Modulators and Evaluating Pi-Terms)

Wine quality is highly dependent and affected by the concentration of chemical compounds present in the wine bottle. Keeping this in mind, we performed all simulations using the real available data for each bottle. After selecting essential modulators, we formed dimensionless groups using all the variables associated with these modulators.

We worked with five essential modulators for chemical data, termed as:$$M_{1}\left(\pi_{1}\right)\approx \mathrm{Fruity}\;,\;M_{2}\left(\pi_{2}\right)\approx \mathrm{Herbal},M_{3}\left(\pi_{3}\right)\approx \mathrm{Floral},\;M_{4}\left(\pi_{4}\right)\approx \mathrm{Woody},M_{5}\left(\pi_{5}\right)\approx \mathrm{Spicy}\;\mathrm{and}\;\mathrm{others}$$

Each modulator $M_{i};\;i=1,2,\ldots \ldots ,5$; is linked with three independent chemical compounds $x_{j},\;j=1,2,3$. The value of each $x_{j}$ has a dimension. For example: the values of chemical compounds for chemical data are given in micrograms per litre. Treating each $x_{j}$ as equally important, we needed to establish a dimensionless functional relationship among those variables. There are several possible ways to do this. However, the choice of a dimensionless relationship is not unique, as the Buckingham Pi theorem does not choose the most physically meaningful variables itself. In this case, the researchers had to rely on experts’ judgement and experience. For the sake of simplicity and understanding, we assumed the three most significant chemical compounds in each module as having different concentration values (in micrograms per litre). To form a dimensionless group, and to calculate the cumulative effect of all variables, any two variables were combined together with respect to a third one by assigning a weight to these variables. We ended up with a large set of possible combinations, which were then used to determine the quality proxy indices. For example: Let us understand the formation of these combinations for variables $x_{j},\;j=1,2,3$. A few functional relationships using these variables are shown below:$$\frac{\sum_{j=1,2}x_{j}}{x_{3}},\frac{\sum_{j=2,3}x_{j}}{x_{1}},\frac{\sum_{j=1,3}x_{j}}{x_{2}},\frac{x_{3}}{\sum_{j=1,2}x_{j}},\frac{x_{2}}{\sum_{j=1,3}x_{j}},\frac{x_{1}}{\sum_{j=2,3}x_{j}},\frac{\prod_{j=1,2}x_{j}}{x_{3}^{2}},\;\frac{x_{1}^{2}}{\prod_{j=2,3}x_{j}},\;\frac{\sqrt{\prod_{j=1,3}x_{j}}}{x_{2}}$$

There could be numerous suitable combinations. We figured out all of these possible combinations through the SQL software program (Details can be found in Appendix A). Each of these functional relationships, known as π-terms, are dimensionally homogeneous, interrelated, and a mirror-image of quality proxy values for all modulators.

We obtained five quality proxy values $Q_{p_{i}}$, i=1, 2,…, 5 based on these Pi-values. Each quality proxy is a function of Pi-value; a function of different chemical compounds $x_{j}$, j=1,2,3. Thus, we have:(3)$$Q_{p_{i}}=f\left(\pi_{i}\left(x_{j}\right)\right)\;\forall \;i=1,2,\ldots 5;j=1,2,3$$

Now, the values of $\pi_{i}s$ are representative of the quality proxy index $Q_{P}$. Hence,
(4)$$Q_{P}=Q_{P}\left(Q_{p_{1}},Q_{p_{2}},Q_{p_{3}},Q_{p_{4}},Q_{p_{5}}\right)$$

Or, more specifically,
(5)$$Q_{P}=f\left(\pi_{1},\pi_{2},\pi_{3},\pi_{4}\;,\pi_{5}\right)$$

These quality proxies $Q_{p_{1}},Q_{p_{2}},Q_{p_{3}},Q_{p_{4}},$ and $Q_{p_{5}}$ are related to the dimensionless parameters $\pi_{1},\pi_{2},\pi_{3},\pi_{4}\;,\pi_{5}$. In addition, each of the π values were obtained using important chemical compounds responsible for that flavour. The selection of these chemical compounds is explained in detail in Section 2.2.1.

In addition, it was necessary to assign weights to these π values as the concentration of each chemical compound varies. In addition, the impact of each variable on the π values based on different flavours is different. For assigning different weights in the form of exponents to obtain the best possible combination for the wine quality parameter, the overall wine quality index using Equation (5) is given as:(6)$$Q_{P}\propto \overset{5}{\underset{i=1}{\prod }}\pi_{i}^{n_{i}}$$
(7)$$Q_{P}=k\;\pi_{1}^{n_{1}}\pi_{2}^{n_{2}}\pi_{3}^{n_{3}}\pi_{4}^{n_{4}}\pi_{5}^{n_{5}}$$
where k is the proportionality constant and $n_{1},\;\ldots \ldots ,n_{5}$ are the weights assigned to each group.

Just to specify again that these chemical compounds are not always the same. This model is flexible in the sense of selecting relevant features related to different kinds of studies based on the geographical locations of vineyards, and also depends on the strategy that a researcher develops. We chose these features based on the available data for 18 samples of New Zealand Pinot noir wines.

Now we need to solve Equation (7) to obtain $Q_{P}$. The choice of different exponent weights is a tedious job. Taking the logarithm on both sides of the Equation (7) and solving it for different combinations of $n_{i}$s will serve the purpose; however, practically finding all these possible solutions is not feasible. So, maintaining equal proportionality for each exponent, we chose a common ratio:(8)$$\frac{n_{2}}{n_{1}}=\frac{n_{3}}{n_{2}}=\frac{n_{4}}{n_{3}}=\frac{n_{5}}{n_{4}}=\alpha$$
i.e.,
(9)$$n_{2}=\alpha n_{1}=\alpha n,\;n_{3}=\alpha n_{2}=\alpha^{2}n,\;n_{4}=\alpha n_{3}=\alpha^{3}n,\;n_{5}=\alpha n_{4}=\alpha^{4}n$$

In this way, we were able to maintain homogeneity by choosing different combinations of α and n, only. Here, we assumed the proportionality constant k as unity for simplicity. We used this k value as a calibration factor to obtain the desired level of quality values of a wine. Further, while DA does not provide information about how to choose the best functional relationship, it guarantees that the final model is comparable and dimensionally homogeneous.

Based on numerous combinations (198, 288) of α and n values, we chose five of the most relevant combinations closer to the experimental results based on Euclidean distance. The pseudocode for the optimization algorithm is explained in the Appendix A. The quality proxy values for Pinot noir wine bottles are shown in Table 6 and plotted in Figure 6. The developed mathematical framework provides lots of flexibility in terms of adjusting the weight of the concentration of the available chemical compounds in the wine bottle, and also in terms of selecting essential modulators based on the type of different vineyards.

### 3.3. Mathematical Results for Case Study 1 and Case Study 2

We did two case studies due to the unavailability of chemical concentration data for data3, and to validate the mathematical framework; case study 1 for physiochemical data for data2 (for which chemical concentration is also available and wine quality has been predicted, as shown in Figure 5) and case study 2 for physiochemical data for data3. The values of different parameters are summarised in Appendix A. For these case studies, we formed two Pi-groups only, as the parameters related to physiochemical data are small. The first Pi-group is formed with total sulphur, total phenolics and titratable acidity, whereas the second Pi-group contains sugar, ethanol, and pH values. The selection of these two separate groups is based on their characteristics and contribution in making a good quality wine. The titratable acidity (TA) is directly related to the functional characteristics, is expressed as perceived sourness and to some degree, astringency, influences overall balance and wine structure, and stabilises the colour. Free sulphur dioxide works as an antioxidant and an anti-microbial. It stops bacteria and unwanted wild yeasts from growing in grape juice and wine. Total phenolics provides an overall measure of phenolic extraction, including coloured and non-coloured phenolics. Phenolics are chemical substances that alter wine’s flavour, texture, and colour. In the second Pi-group, ethanol can promote the perception of hotness and viscosity of wine, and although it gives a perceived sweetness to the wine, it can give a burning sensation in higher concentration. The presence of alcohol with sugars, acids, and phenols (especially tannins) is also important to the overall balance of a wine. An appropriate pH value is also highly desirable in Pinot noir wine, in terms of both sensory phenomena (e.g., perceived wine balance) and physiochemical aspects, including colour stability and microbial safety. Higher pH values are in general associated with lower acidity, and although low acid wine can feel smoother and rounder on the palate, it can appear ‘flabby’ when the acidity is not well balanced with other wine qualities. On the other hand, a wine will be perceived as being sourer and crisper as the pH value drops.

Wine quality values have been predicted based on different sets of parameters for these two Pi-groups. The results are plotted with experimental results obtained via sensory experiments for data2 (Figure 7) and data3 (Figure 8).

A similar set of parameters was used for case study 2, but with different values of parameters. The experimental results do not have a description of the chemical compounds and their concentrations for this dataset. We therefore rely only on the available sensory results and on the physiochemical data.

### 3.4. Validation/Significance of ML Models

After conducting an analysis using DNN on all three datasets (data1, data2 and data3) we predicted the wine quality of 18 bottles, and compared them with wine quality from the experimental results, as mentioned in Figure 9 and Figure 10. We are comparing the predicted wine quality from data1 and data2 together, because these data are from Experiment 1 and are related to the same wines. Figure 9 represents the wine quality predicted via DNN using data1 and data2, based on MAE as the evaluation metrics. According to the results, DNN achieved an MAE of 0.44 on testing the dataset, while predicting the wine quality with data1, which is associated with the chemical features of the wines. Moreover, an appropriate wine quality was achieved with only 278 epochs out of 2000. After comparing the predicted wine quality via DNN using data1 with wine quality from the experimental results, similar trends were noticed in most of the wines, except for OFRCP16, WCR16 and NS16. The wine quality predictions of these three wines via DNN were found to demonstrate noticeable differences when compared with the wine quality judgments by human experts. On the other hand, in the case of data2, which is related to the physiochemical features, DNN predicted wine quality with an MAE of 0.38, which is better than the DNN in data1. However, it took more computational power and epochs to obtain the appropriate results. In addition to that, wine quality prediction via DNN using data2 in general showed similar patterns to those of the predicted wine quality with data1, with one difference being that the wine quality in the case of wines WCR16 and OFRCP16 showed some improvement. It is interesting to note that these three wines that showed greater differences between the empirically observed and predicted quality scores were wines at the endpoints of the perceived wine quality spectrum. That is, wines WCR16 and OFRCP16 received high ratings of quality relative to the other wines in the 18-wine sample set from the expert judges, whilst wine NS16 was consistently judged lowest in quality out of the 18 wines [6].

While predicting the wine quality of wine using data3, which is associated with the physiochemical data from Experiment 3, DNN scored an MAE of 0.35 on testing the dataset at 998 epochs. The predicted wine quality was in agreement with the wine quality judgments from the experimental results, except for wines WTT and WCR, as shown in Figure 10.

When comparing the performance of DNN on three different datasets (data1, data2 and data3), the impact of the input dimension on wine quality prediction was clearly demonstrated. In the case of data1, which contains 14 features/input dimension, MAE was higher but computationally inexpensive. On the other hand, both data2 and data3 with an input dimension of 6 provided better results than data1 in terms of MAE. However, DNN with data2 and 3 were found to be more computationally expensive than DNN with data1. In addition to that, after comparing the performances of DNN models with data2 and data3 related to the physiochemical characteristics of the wines, it was noted that data2 is more complex than data3. The reason for this complexity is the requirement of more nodes for data2 in the first (128) and second layers (128), as compared to the DNN model with data3 (first layer—64, second layer—64).

In terms of the activation function used in all three DNN models, the RELU [57] activation function was common in the first layer of all three DNN models, and ELU was found to be appropriate during DNN with data1 and 3 (from the second to fourth layer). On the other hand, in the case of DNN with data2, an RELU activation function was used in all layers. The activation functions are similar, except that ELU can produce a negative output. Activation functions play a critical role in DNN due to non-linearity in the data. In addition to that, optimisers are also important in the DNN model performance by reducing the error. Optimisers are used to change the attributes of the neural networks such as weight and learning rate, in order to reduce loss. In this study, we utilised RMSprop for DNN with data1, and Adam for DNN with data2 and 3. RMSprop divided the learning rate via an exponentially decaying average of squared gradients. On the other hand, Adam [59] is the combination of RMSprop and stochastic gradient descent with momentum. Overall, based on the DNN results, the use of essential modulators as input can predict wine quality very effectively.

### 3.5. Limitations

There are, however, certain limitations that are associated with this approach. This approach requires prior knowledge of the variables that affect the phenomena, i.e., wine quality, and the choice of these variables determines whether the method is successful or unsuccessful. The selection of useful variables might be a very challenging task. As a result, various lists of the parameters need to be used for description.

One limitation of the Buckingham Pi theorem used for DA is its inability to form unique dimensionless groups. Any basis of the linear subspace of exponent values can be used, and is valid to form dimensionless groups. Another limitation relates to the selection of relevant and important variables that will show their physical relationship with the system. Despite certain limitations, this approach has been used in seminal studies such as Kolmogorov’s turbulence theory [60], Taylor’s estimation of yield in nuclear explosions [61], and many more [62,63,64]. Additionally in this way, this approach provides more flexibility in terms of the testing of important features and to make decisions.

We need to highlight two challenges associated with this approach: The first relates to how a researcher chooses the essential physical quantities, as the theorem is not able to do this. The second challenge relates to the analysis results: the implicit functional relationship is not a complete physical equation and the proportionality constant needs to be determined, either by experimental or theoretical analysis. It then needs to be validated.

## 4. Summary of Findings and Conclusions

Consequent to receiving the perceived wine quality indices from sensory analysis, we developed a mathematical model to predict wine quality, and validated it using machine learning algorithms based on available features, including the chemical and physiochemical factors. To develop the mathematical method, we chose the essential modulators and implemented the Buckingham Pi theorem. As per our knowledge, and after an extensive literature review, we can conclude that the implementation of the mathematical analysis, including the Buckingham Pi theorem to extract essential modulators, together with unsupervised machine learning classification, developed a strong foundation for the development of a predictive model of wine quality. The identification of clusters among chemicals and the extraction of essential modulators, with the help of the literature, played a critical role. Essential modulators were chosen based on their chemical features, such as the odour activity value and the threshold value. We tested several wine quality ratings with the observed values, finding a few to be closer to the original wine quality values rated by experts.

Robustness is one of the required characteristics for any physical model. Dimensional homogeneity is necessary to achieve robustness. DA separates the physical dimensions and the units of measurement [51,65]. For example, while the weight of a chemical compound may vary between a Pinot noir and a Sauvignon Blanc, the physical dimension of that compound remains the same. In short, the units gauging that compound may vary. It is crucial to find critical independent variables (what we called “essential modulators”) to investigate the process in the modelling, simulation and experimentation, and it is also difficult to simplify and to generalise the results. This is especially true in the case of complex problems such as perceived and conceptualised wine quality.

In conclusion, we can say that the mathematical framework provides a solid scientific background for model development. The use of machine learning approaches validated this model and helped to speed up the computing process. Advanced machine learning algorithms provided an effective tool for understanding the complex nature of wine datasets, and conveyed useful information related to wine quality. Based on the proposed approach, we are developing a machine learning-based web application that wine researchers and wine producers can use to forecast wine quality based on some specific features that are present in their wines, and that can be tuned for different variable amounts.

## Figures and Tables

**Figure 1 foods-11-03072-f001:**
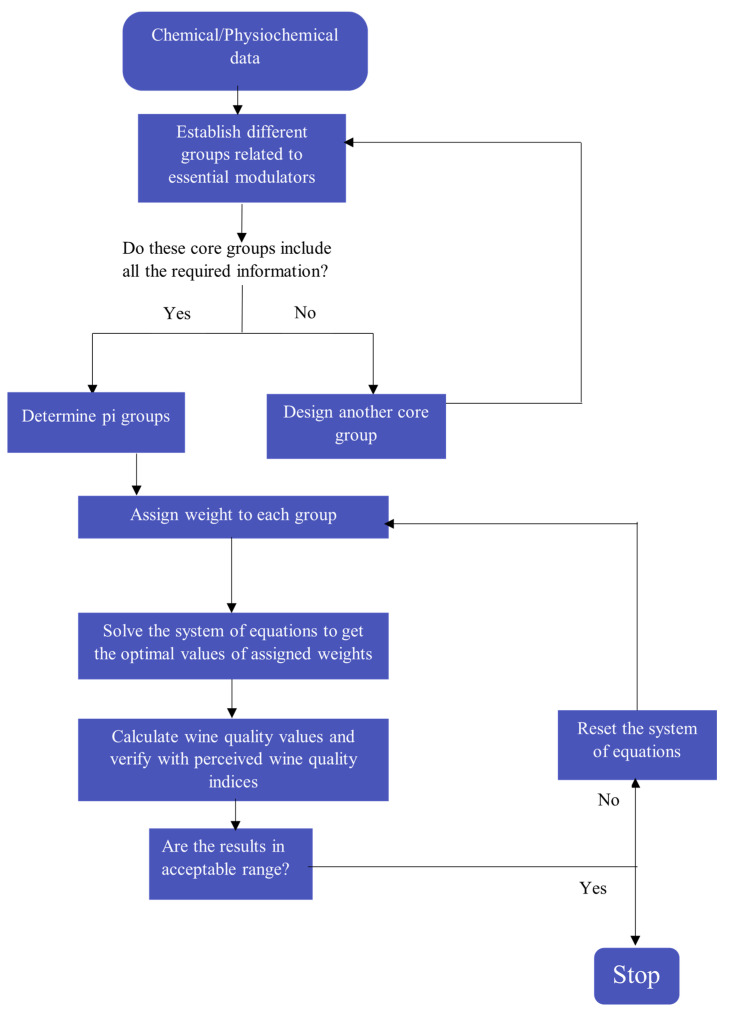
Flow chart for selecting essential modulators for mathematical development of the model.

**Figure 2 foods-11-03072-f002:**
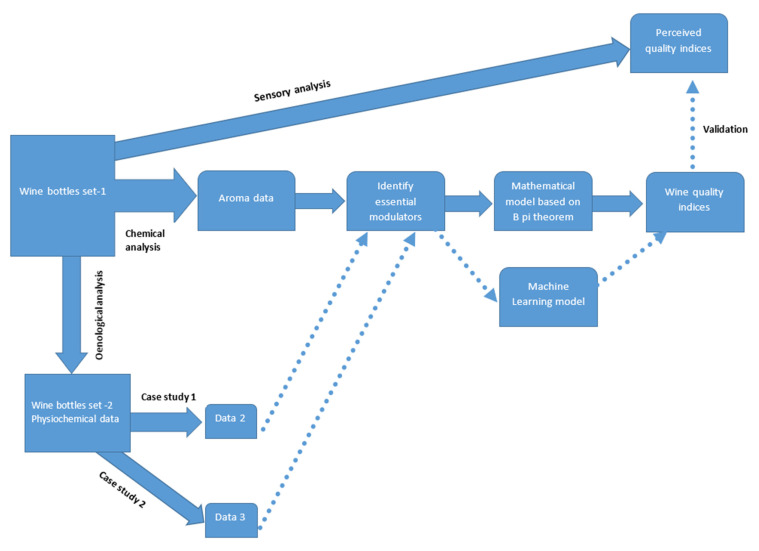
Schematic of developed mathematical framework and validation through two case studies and machine learning model. To produce perceived wine quality indices, a thorough sensory analysis was performed for two 18-wine sets of New Zealand Pinot noir wine (Experiment 1 and Experiment 2) in the first phase, and another set of 18 New Zealand Pinot noir wines (Experiment 3) for the second phase of this research. Based on an extensive literature review and threshold values of different chemical compounds found in the Pinot noir wine bottle, essential modulators are chosen from the available chemical data for set 1 (chemical analysis). These crucial modulators are modelled using the Buckingham Pi theorem in the later stages, and the wine quality indices for bottles of Pinot noir are eventually predicted. These quality indicators are compared to the wine quality indices that were obtained initially through sensory analysis. The model validation phase is completed using machine learning methods.

**Figure 3 foods-11-03072-f003:**
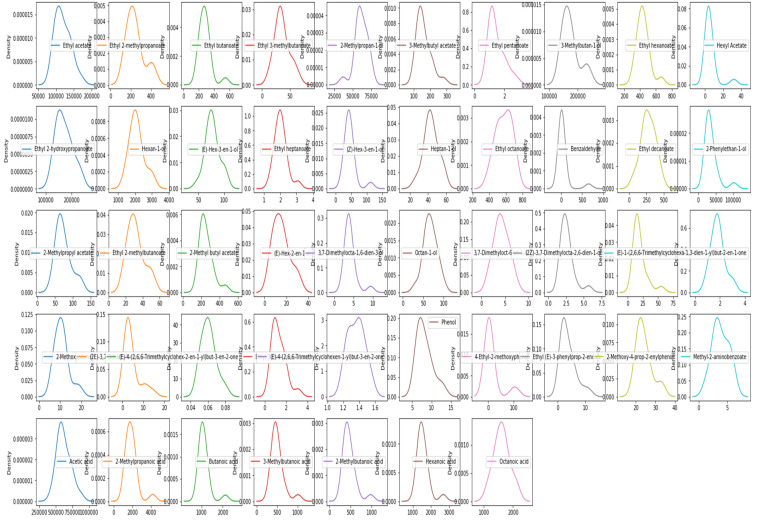
Density plot of each chemical compound present in the New Zealand’s Pinot noir wine dataset1.

**Figure 4 foods-11-03072-f004:**
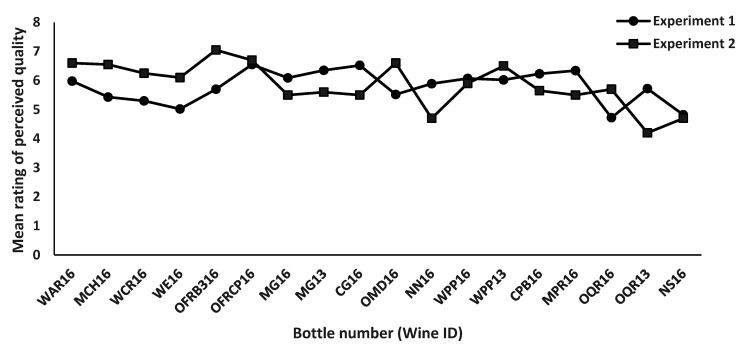
Mean ratings of perceived wine quality obtained in Experiment 1 and Experiment 2 for the same set of 18 wines. Experiment 1 investigated primarily aromatic attributes, whilst Experiment 2 investigated in-mouth attributes of taste and trigeminal effects (mouthfeel).

**Figure 5 foods-11-03072-f005:**
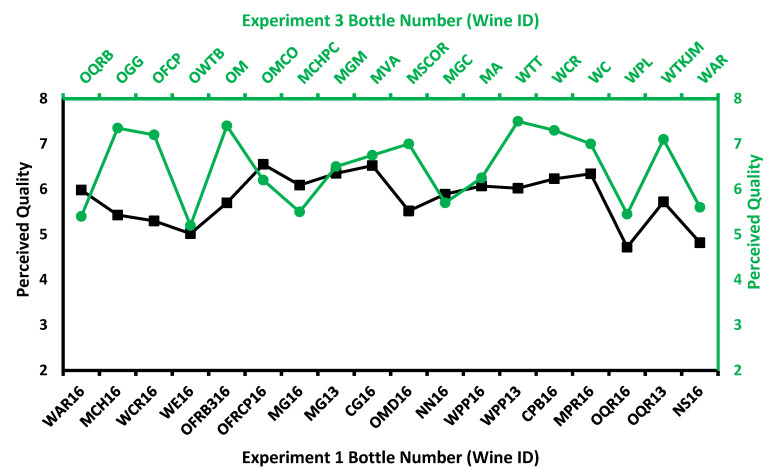
Perceived wine quality ratings for two different sets of New Zealand Pinot noir wine observed in Experiment 1 and Experiment 3.

**Figure 6 foods-11-03072-f006:**
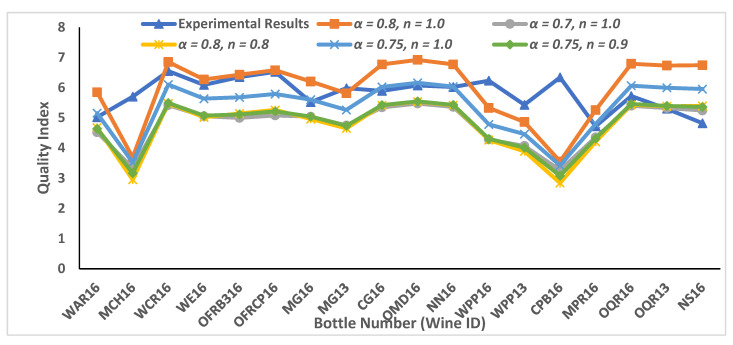
Predicted wine quality values for Pinot noir wines with available experimental results. Five sets of parameters were used for comparative study of available New Zealand Pinot noir wine samples with available sensory results from Experiment 1. Different values of two parameters α and n were used for understanding the impact of different essential modulators on the overall quality value.

**Figure 7 foods-11-03072-f007:**
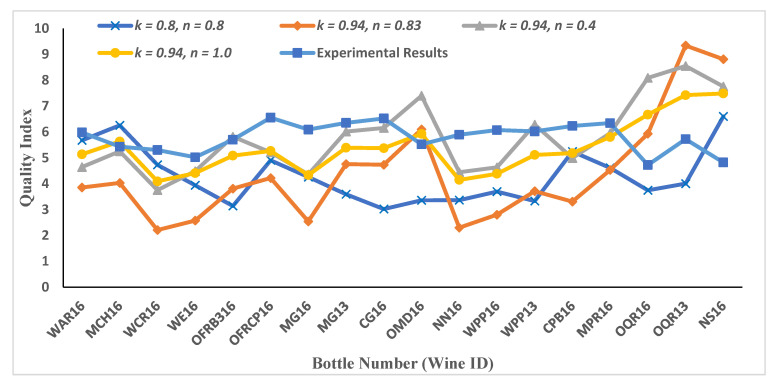
Wine quality results for case study 1 for data2. As explained in Equation (9), we assumed $\frac{n_{2}}{n_{1}}=\alpha =n$. Different values of the parameter k are considered in contrast with the value of k taken as unity in Figure 5. Diverse results are obtained for different combinations of k and n. Results for k=0.94 and n=1.0 are comparable with experimental results, which shows that both Pi-groups have equal importance and weightage, and suggests that there should be a balance between various physiochemical compounds.

**Figure 8 foods-11-03072-f008:**
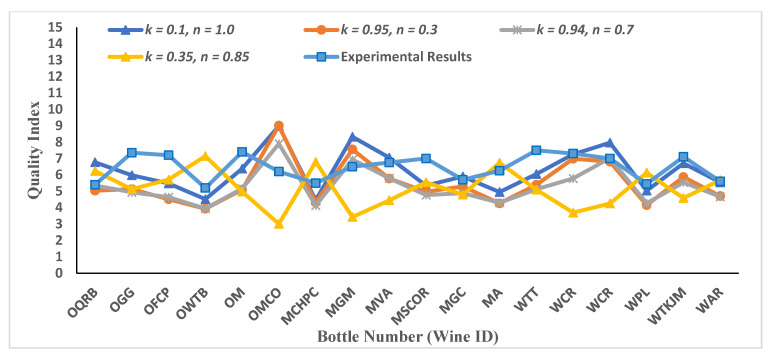
Wine quality results for case study 2 for data3. As explained in Equation (9), we assumed $\frac{n_{2}}{n_{1}}=\alpha =n$. Different values of the parameter k are considered in contrast with the value of k taken as unity in Figure 5. Results obtained in this case study are denser as compared to case study 1. This dataset is quite different as compared with data2, and a few outliers are also there. Though any concrete results cannot be obtained from the predicted values, results are comparable and clearly suggest providing the balance weightage to each Pi-group.

**Figure 9 foods-11-03072-f009:**
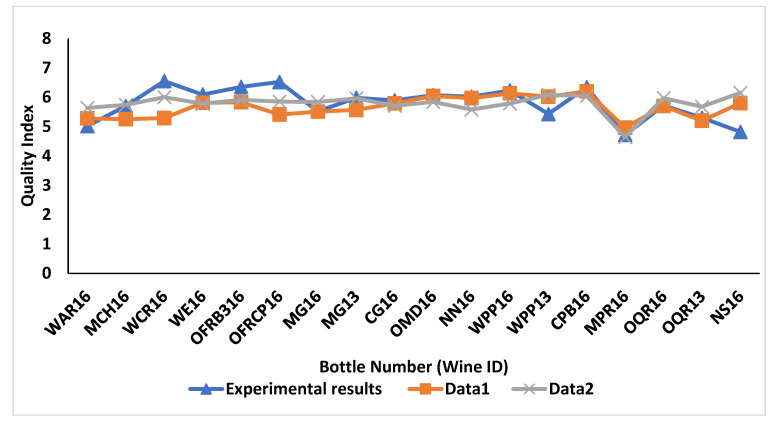
Wine quality prediction for data1 and data2 based on MAE as evaluation metrics using deep neural networks.

**Figure 10 foods-11-03072-f010:**
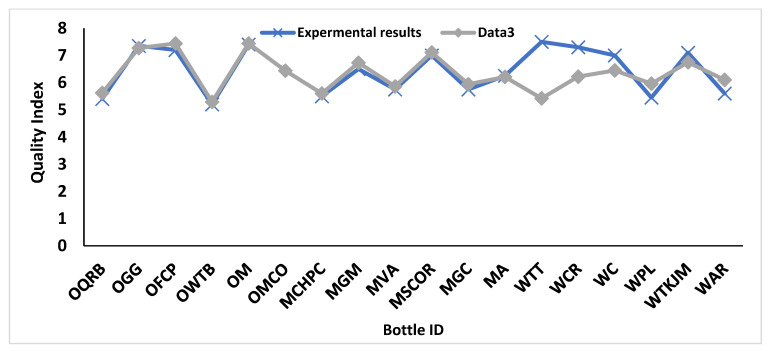
Wine quality prediction for data3.

**Table 1 foods-11-03072-t001:** The 18 wines employed in the two experiments.

Wine Identity	NZ Region	Vintage	Price Point	Vine Yield	Production Philosophy	Closure	RRP NZD
WAR16	Wairarapa	2016	Premium	Low	BioGro cert	SC	82
WPP16	Wairarapa	2016	Commercial	Mod	conventional	SC	26
WPP13	Wairarapa	2013	Commercial	Mod	conventional	SC	26
WE16	Wairarapa	2016	Premium	Low	Organic in transition	SC	52
WCR16	Wairarapa	2016	Premium	Low	conventional	SC	140
MCH16	Marlb	2016	Premium	Low	BioGro cert	Cork	44
MPR16	Marlb	2016	Commercial	High	conventional	SC	15
MG16	Marlb	2016	Premium	Low	BioGro cert	SC	63
MG13	Marlb	2013	Premium	Low	BioGro cert	SC	63
OMD16	Central Otago	2016	Commercial	High	conventional	SC	28
OFRCP16	Central Otago	2016	Premium	Low	Demeter cert	SC	76
OFRB316	Central Otago	2016	Premium	Low	Demeter cert	SC	102
OQR16	Central Otago	2016	Premium	Low	BioGro and Demeter cert	SC	79
OQR13	Central Otago	2013	Premium	Low	BioGro and Demeter cert	SC	79
NN16	Nelson	2016	Premium	Low	BioGro cert	SC	67
NS16	Nelson	2016	Commercial	High	conventional	SC	13
CPB16	Nth Canterbury	2016	Commercial	Mod	conventional	SC	22
CG16	Nth Canterbury	2016	Premium	Low	organic in transition	SC	43

Marlb = Marlborough; SC = screwcap bottle closure; RRP = recommended retail price; Vine yield: low ≤ 2 kg/vine; mod and high ≥ 2 kg/vine. Price point: Premium ≥ NZD 30.00; Commercial ≤ NZD 30.00.

**Table 2 foods-11-03072-t002:** Wine attributes rated in Experiments 1 and 2. The attributes were rated as descriptors on 10-point category scales ranging from 1 to 10 in each sensory procedure. The anchors for each 10-point scale are given.

Descriptors Experiment 1	Anchors	Descriptors Experiment 2	Anchors
Attractive fruit aromatics	Low–Intense	Overall quality	Poor–Good
Attractive floral aromatics	Low–Intense	Softness/gentleness/suppleness	Low–Intense
Earthy/mushroom notes	Low–Intense	Smoothness/silky/velvety	Low–Intense
Reductive notes	Low–Intense	Weight/heaviness/density	Low–Intense
Bitterness	Low–Intense	Volume/fullness/roundness	Low–Intense
Astringency	Low–Intense	Viscosity/mouth-coating/oiliness	Low–Intense
Sweetness	Low–Intense	Bitterness	Low–Intense
Harshness of tannins	Low–Intense	Dry/puckering	Low–Intense
Green/herbaceous notes	Low–Intense	Dusty	Low–Intense
Overall quality	Poor–Good	Coarse/grainy/rough	Low–Intense
Balanced acidity	Poor–Good	Harsh tannins/aggressive	Low–Intense
Elegance/precision	Poor–Good	Astringency	Low–Intense
Softness/silkiness	Poor–Good	Thin/watery	Low–Intense
Freshness	Poor–Good	Burning sensation/hot	Low–Intense
Expressiveness	Poor–Good	Sourness/acidity	Low–Intense
Fruit ripeness	Poor–Good	Overall body	Poor–Good
Oak influence	Poor–Good	Pinot noir varietal typicality	Poor–Good
Concentration in mouth	Poor–Good		
Overall structure	Poor–Good		
Pinot noir varietal typicality	Poor–Good		

**Table 3 foods-11-03072-t003:** List of selected chemical compounds to build up essential modulators.

Essential Modulators	Chemical Compounds
Fruity (π_1_)	Ethyl octanoate
	Ethyl butanoate
	Ethyl hexanoate
Herbal (π_2_)	Hexan-1-ol
	(E)-Hex-3-en-1-ol
	Heptan-1-ol
Floral (π_3_)	(E)-1-(2,6,6-Trimethylcyclohexa-1,3-dien-1-yl) but-2-en-1-one (Beta)
	2-Phenylethan-1-ol
	3,7-Dimethylocta-1,6-dien-3-ol (linalool)
Oak and Woody (π_4_)	4-Ethyl-2-methoxyphenol (eugenol)
	Benzaldehyde
	2-Methoxyphenol (guaiacol)
Others (π_5_)	Phenol
	4-Ethyl-2-methoxyphenol (eugenol)

**Table 4 foods-11-03072-t004:** List of 47 chemical compounds in New Zealand’s Pinot noir wine dataset1.

List of Chemical Compounds	
Ethyl acetate	(E)-Hex-3-en-1-ol
3-Methylbutyl acetate	Ethyl heptanoate
Ethyl pentanoate	(E)-Hex-2-en-1-ol
Ethyl 2-hydroxypropanoate	Octan-1-ol
(Z)-Hex-3-en-1-ol	(E)-1-(2,6,6-Trimethylcyclohexa-1,3-dien-1-yl) but-2-en-1-one
Ethyl octanoate	2-Methoxyphenol
Benzaldehyde	(E)-4-(2,6,6-Trimethylcyclohexen-1-yl) but-3-en-2-one
Ethyl decanoate	2-Methylpropanoic acid
(2E)-3,7-Dimethylocta-2,6-dien-1-ol	Butanoic acid
(E)-4-(2,6,6-Trimethylcyclohex-2-en-1-yl) but-3-en-2-one	Ethyl 2-methylpropanoate
Acetic acid	Hexyl Acetate
2-Methylbutanoic acid	Heptan-1-ol
Ethyl butanoate	2-Methyl butyl acetate
Ethyl 3-methylbutanoate	3,7-Dimethylocta-1,6-dien-3-ol
3-Methylbutan-1-ol	3,7-Dimethyloct-6-en-1-ol
2-Phenylethan-1-ol	methyl-2-aminobenzoate
2-Methylpropyl acetate	2-Methoxy-4-prop-2-enylphenol
Ethyl 2-methylbutanoate	Methyl-2-aminobenzoate
(2Z)-3,7-Dimethylocta-2,6-dien-1-ol	3-Methylbutanoic acid
2-Phenethyl acetate	Hexanoic acid
Phenol	Octanoic acid
4-Ethyl-2-methoxyphenol	2-Methylpropan-1-ol
Ethyl (E)-3-phenylprop-2-enoate	Ethyl hexanoate
	Hexan-1-ol

**Table 5 foods-11-03072-t005:** Number of selected parameters in each dataset.

Parameters	Data1	Data2	Data3
Input dimension	14	6	6
Layers	5	5	5
Node (layer1)/Activation	64/RELU	64/RELU	128/RELU
Node (layer2)/Activation	64/ELU	64/RELU	128/ELU
Node (layer3)/Activation	32/ELU	64/RELU	64/ELU
Node (layer4)/Activation	16/ELU	8/RELU	64/ELU
Node (layer 5)	1	1	1
Optimiser	RMSprop	Adam	Adam
Epoch with early stopping	2000	2000	2000
Loss metric	MAE	MAE	MAE

**Table 6 foods-11-03072-t006:** Quality proxy indices based on a mathematical model.

Wine ID Set A	Perceived Wine Quality	Wine Quality Proxy Indices (Mathematical Model)				
		*α* = 0.8 *n* = 1.0	*α* = 0.7 *n* = 1.0	*α* = 0.8 *n* = 0.8	*α* = 0.75 *n* = 1.0	*α* = 0.75 *n* = 0.9
WAR16	5.02	5.848	4.521	4.678	5.154	4.639
MCH16	5.7	3.688	3.333	2.950	3.520	3.168
WCR16	6.55	6.849	5.418	5.479	6.101	5.491
WE16	6.09	6.271	5.051	5.017	5.636	5.072
OFRB316	6.35	6.428	4.997	5.142	5.678	5.110
OFRCP16	6.52	6.576	5.077	5.261	5.790	5.211
MG16	5.52	6.203	5.039	4.962	5.601	5.041
MG13	5.98	5.813	4.748	4.651	5.263	4.736
CG16	5.89	6.768	5.346	5.415	6.025	5.423
OMD16	6.07	6.920	5.471	5.536	6.163	5.546
NN16	6.02	6.768	5.360	5.415	6.032	5.429
WPP16	6.23	5.327	4.278	4.262	4.782	4.304
WPP13	5.43	4.863	4.071	3.890	4.459	4.013
CPB16	6.34	3.554	3.267	2.843	3.421	3.079
MPR16	4.72	5.257	4.353	4.205	4.792	4.313
OQR16	5.72	6.790	5.398	5.432	6.065	5.458
OQR13	5.3	6.730	5.322	5.384	5.995	5.396
NS16	4.82	6.742	5.242	5.394	5.954	5.359

## Data Availability

The datasets generated during or analyzed during the current study are available from the corresponding author on reasonable request.

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
