# Peer review of "Understanding Quality of Pinot Noir Wine: Can Modelling and Machine Learning Pave the Way?"

_foods, 2022, doi:10.3390/foods11193072_

Round 1
Reviewer 1 Report
The topic of the submitted article “Understanding quality of Pinot Noir wine: can modelling and machine learning pave the way?” is of interest for the readers and have interesting findings. Please see the comments below.
Please clearly present the hypothesis and contribution of your study in the Introduction section.
Line 8-14: Theses lines can be more concise. The abstract should ideally review specific objectives, materials, results, conclusions, and applications as concisely as possible.
Line 37: Why was pinot noir considered for machine learning approaches, explain further in introduction the industrial relevance of this approach
Line 111: provide reference and what’s been conveyed here
Line 122-126: Try to keep the average sentence length around 20-25 words.
Line 140: The review of the literature overall could have been better so the reader is given an adequate background about the topic.
Line 170: Font seems to be different, check throughout the manuscript
Line 203: Was any other approach considered, more explanation anticipated
Line 216: How was this sample size of 18 deemed enough?
Line 230: Why was this characterization data missing
Line 237: Why was this approach of simulated synthetic data used
Line 268: Tables and figures must be standalone, consider avoiding abbreviations or have it in footnotes.
Line 272: Not clear; more clarification required on how this experimental design was made
Line 284: Was the time of sensory sessions kept consistent
Line 293: On what basis were the participants selected and what were the background information shared to them regarding the study
Line 333: If a different panel was employed what is the expected repeatability of this approach
Line 357: Explain further how these parameters are critical. Were any preliminary studies carried out?
Line 441, 452: Check the font and spacing.
Line 560: Include the SD on all the possible graphs
Line 795-799: further explanation anticipated
Line 809: Include a comment indicating the potential use of the proposed approach at an industrial level.
Reviewer 2 Report
Very interesting, well written and novel work on the possibility of modelling the consumers perception of wine quality.
The only few comments I would like to do, and expect the authors to take it into consideration, are:
- - What future developments are expected?
- - Is it designed for commercial use?
- - Can the developed model be used for all wine (other varieties, countries, etc.) or does it need to be validated in other datasets?
Round 2
Reviewer 1 Report
Do not require any further clarification, thank you.